# Lumpy skin disease outbreaks in Egypt during 2017-2018 among sheeppox vaccinated cattle: Epidemiological, pathological, and molecular findings

**Sherin R. Rouby**[1]*, **Nesreen M. Safwat**[2], **Khaled H. Hussein**[1], **Aml M. Abdel- Ra'ouf**[3], **Bahaa S. Madkour**[3], **Ahmed S. Abdel-Moneim**[4], **Hosein I. Hosein**[1]

1 Department of Veterinary Medicine, Faculty of Veterinary Medicine, Beni-Suef University, Beni Suef, Egypt,
2 Department of Pathology, Faculty of Veterinary Medicine, Beni-Suef University, Beni Suef, Egypt,
3 Department of Animal Medicine, Faculty of Veterinary Medicine, Aswan University, Aswan, Egypt,
4 Microbiology Department, College of Medicine, Taif University, Al-Taif, Saudi Arabia

* shereen.rouby@vet.bsu.edu.eg

**Data Availability Statement:** All relevant data are within the manuscript and its Supporting

## Abstract

The General Organization of the Veterinary Services in Egypt has adopted a sheeppox vaccination policy to control lumpy skin disease (LSD) in cattle. Over the course of the last two years, recurrent outbreaks were reported, with animals showing severe clinical signs and consequentially higher fatalities than that of cases reported in previous LSD outbreaks. A total of 1050 cattle showing typical clinical signs suggestive of LSD were clinically and pathologically investigated during 2017–2018. Skin nodules were collected and lumpy skin disease virus (LSDV) was screened in collected skin samples using PCR for the RPO-30 gene. Furthermore, the entire P32 protein coding gene was sequenced. Histopathology and immunohistochemistry of the skin nodules were also conducted. The obtained results showed an overall mortality rate of 6.86%. LSDV was confirmed in all the examined nodules as evidenced by immunohistochemistry and positive PCR amplification of the RPO30 gene. Sequencing analysis of the P32 gene revealed a highly conserved nature and genetic stability of the LSDV. The results of the present study show that the current vaccination protocol was not effective for a multitude of reasons. These results also serve as evidence for a strong recommendation of an amendment of homologous vaccine use aside from a complete coverage of cattle populations in order to reduce the incidence of LSD among cattle population in Egypt.

## Introduction

LSD is a devastating, high-impact cattle disease. It is characterized by fever and nodules covering all parts of the body, including the mucosal membranes and internal organs, along with generalized lymphadenopathy. It also induces, abortion, damage to hides and a sharp decline

information files. Accession numbers denote nucleotide sequence of LSD P32 gene and all are available in the NCBI database by using the Accession numbers MN418201 MN418202 MN418200.

**Funding:** The authors acknowledge the support of Taif University Researchers Supporting Project Number (TURSP-2020/11), Taif University, Taif 21944, Saudi Arabia.

**Competing interests:** The authors have declared that no competing interests exist.

in milk production; it also causes sterility in bulls, subsequently leading to drastic economic losses [1].

LSD is caused by a lumpy skin disease virus (LSDV) that is related to the genus *Capripox-virus* [2]. Blood-sucking arthropods, mosquitoes and ticks transmit the virus from one animal to another [3].

With its emergence in Egypt in 1988, LSD was seen merely as a typical example of an exotic disease that usually entered the country through importation of live animals. Within a few short years, it transformed into a national crisis with a subsequent establishment of the disease's enzootic status. The disease then re-emerged in the summer of 1989 [4]. In 2006, LSDV struck most of the governorates in Egypt [5] and it re-appeared again in 2011 and 2014 [6]. According to this study, and in addition to evidence from the above-mentioned reports, it can be seen that the disease continues to spread across the country.

Originally, the disease affected cattle within the continent of Africa. However, since 2012, the disease has spread from these African countries to the Middle East and even some European countries [7]. Outbreaks of LSDV typically occur as epidemics [4]. There are often unexplained gaps of several years between different outbreaks, regardless of humidity, season or abundance of the vectors [8].

LSDV, sheeppox virus (SPPV), and goatpox virus (GTPV) showed genome similarities of at least 96% [9, 10]. For this reason, conventional laboratory tests could not differentiate different capripoxviruses from each other [11]. However, various PCR assays have been recently validated for differentiation between LSDV and SPPV. PRO-30 based PCR provides a simple and quick differentiation between LSDV and SPPV without the requisite of DNA sequencing [12]. The Capripoxvirus P32 gene is a major immunodominant antigen located at the surface of the mature virion [13, 14]. It was identified as the homolog of the vaccinia virus H3L gene. On the other hand, the RPO30 gene is a homologue of the vaccinia virus E4L gene, which encodes DNA-dependent RNA polymerase enzyme [12]. The main objective of this study was to highlight the vaccination failure associated with the spread of the disease in Egypt. Among the objectives of the current study was the nucleotide sequence relatedness screening between the published reference strains and SPPV vaccines and the virus strains detected in this study.

## Material and methods

### Ethical approval

The animal ethical committee of the Faculty of Veterinary Medicine, Beni-Suef University, Egypt, approved the present study. Clinical samples used in this study were collected after approval of all the owners' of study animals.

### Animals

A total of 1050 of cattle clinically affected with LSD were screened during 2017–2018 from the Beni Suef, Sohag and Aswan Governorates. Diseased animals ranged from the age of 3 months to 5 years and included individually reared cases and cases from dairy farms (Table 1). All affected animals were vaccinated within the national annual vaccination program with the Romanian SPPV vaccine ($10^3$ TCID50, Veterinary Serum and Vaccine Research Institute [VSVRI], Egypt). The development of the disease was reported three months after vaccination. Most of the examined animals were reared near different watercourses, ditches and gullies, many of which were found to harbour hard ticks.

**Table 1. Number of clinically diseased animals included in the study.**

| Animal | Number | Average age |
|---|---|---|
| Pregnant cows | 320 | 3–5 years |
| Lactating non pregnant cows | 570 | 3–5 years |
| Heifers | 82 | 8–13 months |
| Calves | 78 | 3–12 months |
| Total | 1050 | |

## Samples

Ten skin nodules were aseptically collected from LSDV-infected cattle showing a severe form of the disease. A local subcutaneous infiltration of lidocaine HCl 2% (1ml / 1cm$^2$ of skin) was applied, followed by a small surgical incision of about 0.5 cm x 0.5 cm in diameter which was made to obtain skin samples. Skin nodules were bisected into two equal halves. The first one was suspended in a sterile phosphate buffer saline (pH 7.2) containing gentamicin (50 μg/ml) in the ice box for PCR assays. The other part was placed in a neutral buffered formalin solution of 10% for further histopathological procedures.

## Virus detection

**PCR.**   Extraction of viral DNA was performed on the nodule homogenate using the gSync™ DNA extraction kit, Geneaid (Taiwan) according to the manufacturer's instructions. PCR runs were performed using PCR master mix (Thermo, Germany) in a total volume of 25μl/reaction. A primer set of the RPO-30 gene that flanks 172bp and 151bp for LSDV and SPPV was used according to Lamien *et al.* (2011; Table 2). The reaction parameters were denaturation at 94˚C for 5 minutes, 35 cycles at 94˚C for 45 seconds, 55˚ C for one minute and 72˚ C for one minute, with a final extension of 7 minutes. PCR products were visualized with a transilluminator after being electrophoresed in 1.5% agarose gel. Lyophilized sheeppox vaccine virus (Romanian strain) and nuclease-free water were used as positive and negative controls, respectively.

**Immunohistochemistry.**   LSD suspected skin samples were subjected to immunohistochemistry staining using Dako Cytomation (Denmark A/S (LSAB+ System-AP)—Edition 06/07—Code K0678). Sections were washed with Tris buffer solution, and the endogenous alkaline phosphatase (AP) was blocked by using 1 mM levamisole. Sections were incubated at room temperature for 30 minutes with normal goat serum to block the non-specific binding immunoglobulins. This was followed by incubation with primary rabbit anti-LSDV polyclonal hyperimmune serum, which was locally prepared and used at a dilution of 1:900 for 1 hour at room temperature. Subsequently, samples were further incubated with biotinylated anti-rabbit monoclonal antibodies and streptavidin AP for 30 minutes each at room temperature. AP substrate was then added for three minutes at room temperature.

**Table 2. Sequence of LSD oligonucleotide primers.**

| | Primer | *Tm* | Reference |
|---|---|---|---|
| **RPO30** | F 5'-TCTATGTCTTGATATGTGGTGGTAG-3' | 55˚C | (Lamien et al., 2011) |
| | R 5'-AGTGATTAGGTGGTGTATTATTTTCC-3' | | |
| **P32** | F 5'- ATG GCAGAT ATC CCA TTA-3' | 50˚C | (Heine et al., 1999) |
| | R 5' ACTCTCATTGGTGTTCGG-3' | | |

**Full length sequence of the P32 gene.** A PCR for a whole P32 protein coding gene was employed using previously designed primers (Table 2) to confirm the presence of LSDV DNA in all skin samples (n:10) as well as in two archival samples from the 2012 Egypt outbreak; this formed a total of 12 samples [14]. An initial denaturation at 94˚C for 5 minutes was followed by 35 cycles at 94˚C for 45 seconds, 50˚C for one minute and 72˚C for one minute, with a final extension of 7 minutes. PCR amplicons of the expected size were excised from the gel and extracted using a Geneaid gel extraction kit (New Taipei City, 22180 Taiwan, China) as according to the manufacturer's instruction. Purified PCR amplicons were sequenced commercially. Raw sequences were processed using MEGA X [15]. BLAST analysis was initially implemented to establish sequence identity to GenBank accessions. Evolutionary analyses and sequence alignments were conducted using MEGA X software [15]. A tree was generated by the maximum likelihood method with 1,000 bootstrapped data sets. The Kimura 2-parameter was used as a model and the tree was obtained initially by Neighbor-Join and BioNJ algorithms [16]. The maximum composite likelihood (MCL) was estimated as a matrix of pairwise distances.

**Histopathological examination.** Different tissue specimens were paraffin embedded, fixed in neutral buffered formalin solution 10%; and sectioned at 5–7µm; the specimens were then stained with hematoxylin and eosin according to [17].

## Results

### Clinical examination

Cattle (n: 1050) from three different governorates (Fig 1) were clinically examined. The animals all showed typical LSD clinical signs, such as nodules on the limbs, head, neck and/or trunk, fever (39.5˚C-41.0˚C), lymphadenopathy and oedema in the brisket, perineum,

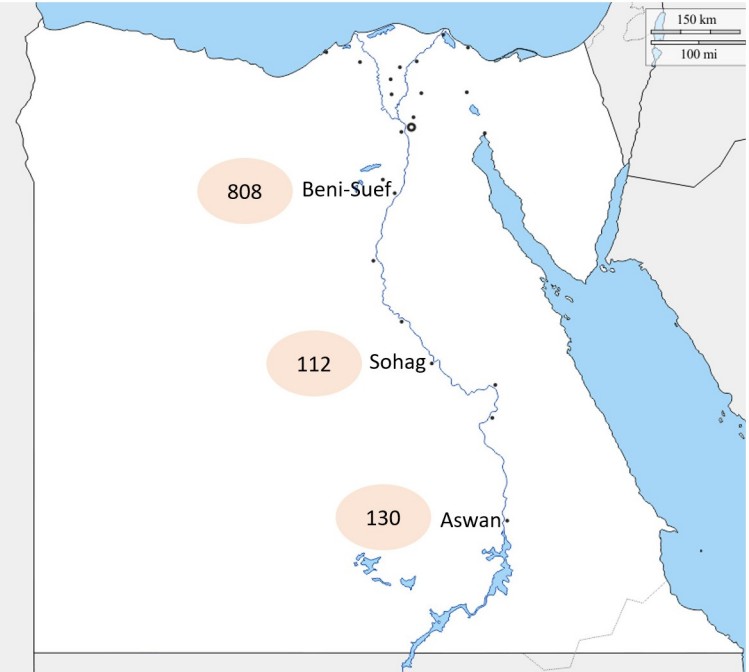

**Fig 1. Areas of investigation and number of animals.** In Beni Suef, 808 animals showed clinical signs of LSD. Twelve calves (3–5 months) and ten pregnant cows were succumbed from infection. In Sohag, 112 animals showed clinical signs of LSD and eighteen cows died. In Aswan, 130 animals showed clinical signs of LSD and thirty-two cows died.

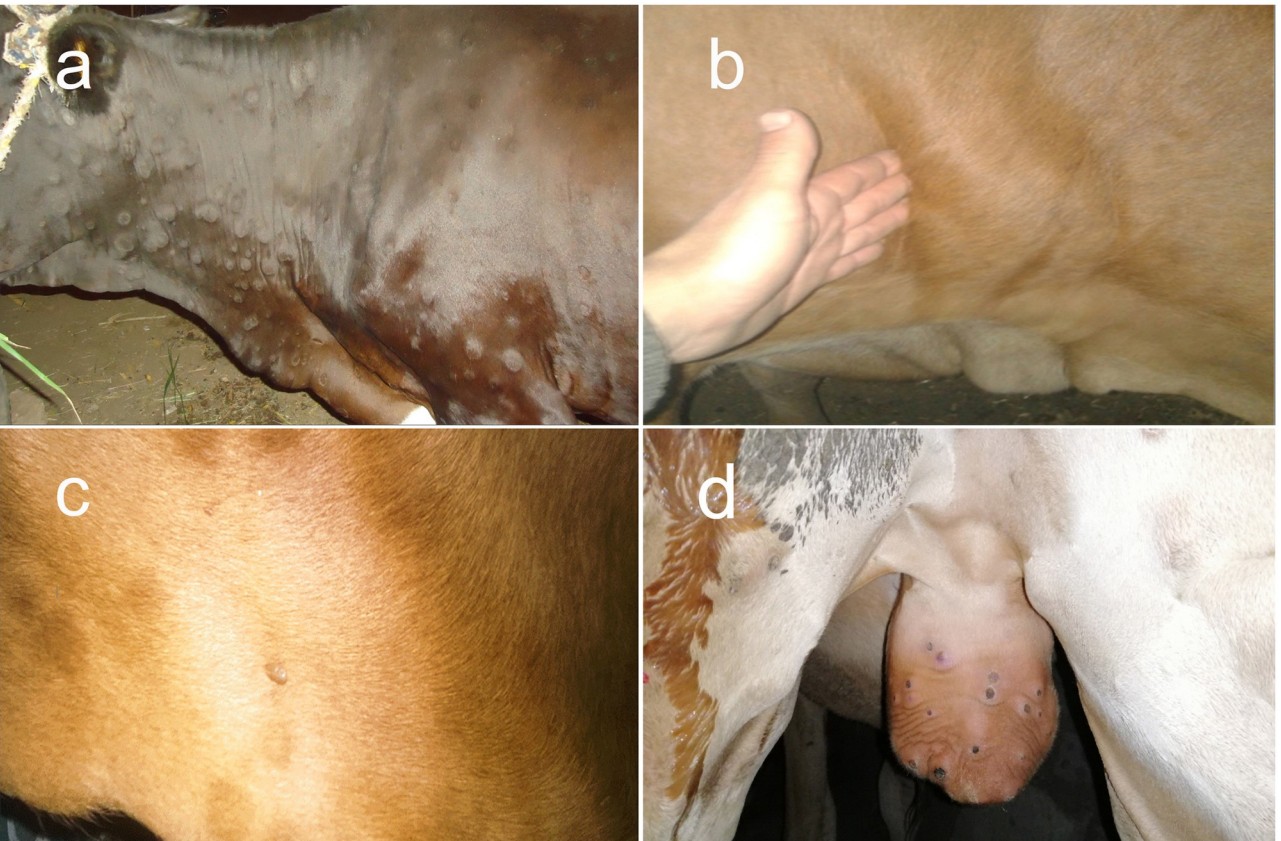

**Fig 2. LSDV infected animals.** A) Cutaneous nodules covered the entire body of a LSDV infected cow. B) Enlarged and oedematous pre-scapular lymph node in a LSDV infected cow. C) LSDV infected cow infested with hard ticks. D) A 12-month-old calf with scrotal lesions. Each lesion is surrounded by a zone of necrosis (sitfast).

genitalia, udder, belly or legs. Typical necrotic lesions with a cone shaped core (sit- fast lesion) were observed (Fig 2). A total of 72 (6.86%) animals succumbed from the acute signs of the disease (Table 3).

## Virus detection: PCR and immunohistochemistry

LSDV DNA was successfully amplified by RPO-30 PCR in all of the 10 selected samples as well as for the positive control (Fig 3). Furthermore, the immunohistochemistry confirmed the presence of a viral antigen, as can be seen in the Fig 4g.

**Table 3. Clinical and epidemiological status of LSDV infected animals.**

| Locality | Number of clinical cases | Mortalities (%) | Complications (%) |
|---|---|---|---|
| Beni Suef | 808 | 22 (2.72%) | Pneumonia 44(5.44%) |
| Sohag | 112 | 18 (16.07%) | Enteritis 24(21.43%) |
| Aswan | 130 | 32 (24.62%) | Bloody urine 22(16.92%) |
| Total | 1050 | 72 (6.86%) | Total 90 (8.57%) |

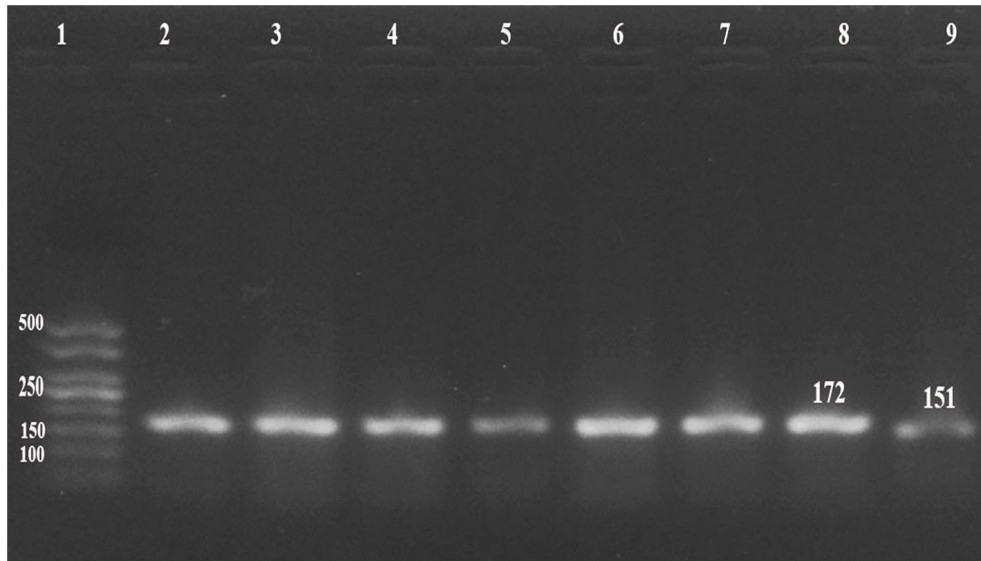

**Fig 3. Gel electrophoresis of PCR products.** Gel electrophoresis of PCR products using *RPO30* specific primer set using 50bp ladder (with 172 bp expected product for LSDV (Lane 2 to lane 8) and 151 for SPPV (Lane: 9).

## Sequence analysis

Four out of the ten 2017–2018 samples and two out of the two 2012 samples were positive for P32 gene amplification (1024bp). Out of the twelve samples, only three showed successful readable sequences (one from 2018 and two from 2012). The processed gene sequences were compared with other P32 sequences of capripoxviruses available in the GenBank. Sequence analysis revealed that LSDV obtained in the current study shared 100% identity on both nucleotide and amino acid sequences with LSDV/NI-2490 (AF325528.1) isolated from cattle in 1958. However, nucleotide and amino acid identities were found to be 99% and 98% with the currently used sheeppox vaccinal strain, respectively (Romanian strain) (data not shown). Phylogenetic analysis of capripoxviruses isolated from 1958 to 2018 revealed that members of the genus *Capripoxvirus* could be classified into three distinct clusters of LSDV, GTPV and SPPV based on their P32 gene nucleotide sequence. LSDV sequences obtained in the present study clustered along with: LSDV/NI-2490/1958 (AF325528.1), LSDV/NW-LW/South Africa/1999 (AF409137.1), LSDV/Kenya/KSGP-0240/1974 (KY702007.1), LSDV/Israel/2012 (KX894508.1), LSDV/Egy/BSU/2012 (MN418201), LSDV/Egy/BSU/2012-2 (MN418202), LSDV/Russia/Dagestan/ 2015 (MH893760.2), LSDV/1015/Egy/2015 (KU298638.1), LSD-E-gypt/Ismailia/2016 (KX977487.1) and LSDV/Iran/2016 (KX960780.1) (Fig 5). Deduced amino acid sequences revealed that amino acid residues at position 49(F/L) and 304(D/N) are unique to LSDV while 62(L/F), 132(S/L) and 134(I/T) are unique to SPPV. The most obvious difference between LSDV and SPPV is the presence of an additional aspartic acid at the 55th position of the P32 of the sheeppox virus; the conserved cysteine residues are present at positions 85 and 89 of SPPV protein sequences and at 84 and 88 in LSDV and GTPV protein sequences (Fig 6).

## Histopathological examination

Biopsy of LSD skin nodules showed acanthosis and vacuolar degeneration in the keratinocytes of the epidermal layer (Fig 4a). Some skin nodules showed dilatation of lymph vessels (LVs),

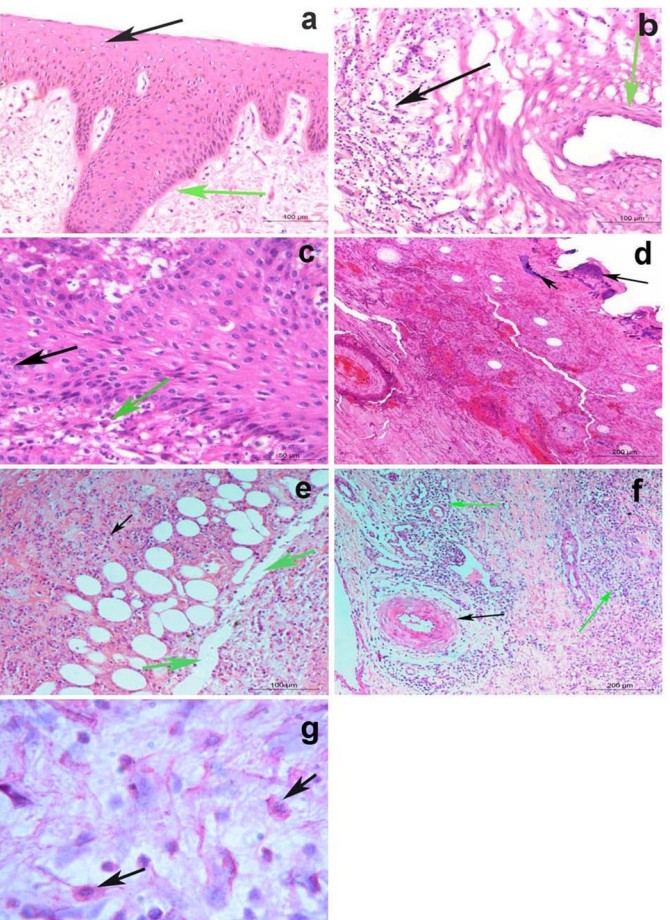

**Fig 4. Histopathology of skin nodules.** a) Acanthosis (green arrow), associated with hydropic degeneration in keratinocytes of the epidermal layer of the skin (black arrow), while the next dermal layer is normal (H&E; Bar = 100μm). b) Vasculitis (green arrow), associated with massive leucocytic infiltrations in the dermal layer (black arrow) (H&E; Bar = 100μm). c) Intracytoplasmic inclusion body of LSD in the keratinocytes of the epidermal layer (black arrow), while the dermal layer shows massive leucocytic infiltrations mainly by eosinophils (green arrow) (H&E; Bar = 50μm). d) Chromatin condensation and calcium deposits in keratin layer (black arrows), while the next dermal layer shows severe hemorrhages with leucocytic aggregation (H&E; Bar = 200μm). e) Necrosis and dissociation of muscle fibers, fatty infiltrates, and leucocytic infiltration mainly by lymphocytes and few numbers of eosinophils between them (black arrows) (H&E; Bar = 100 μm). f) Dilatation of lymph vessels (LVs.), necrotizing vasculitis (black arrow), and leucocytic infiltration mainly by lymphocytes and few numbers of eosinophils between them (green arrows) (H&E; Bar = 200 μm). g) Red viral particles in macrophage of connective tissue of the dermal layer (black arrows) using alkaline phosphatase immunohistochemistry.

vasculitis and leucocytic infiltration, primarily by lymphocytes (Fig 4b). Intracytoplasmic inclusion bodies were observed in the epidermal layer, while the dermal layer showed massive leucocytic infiltration primarily by eosinophils (Fig 4c). Chromatin condensation and calcium deposits in the keratin layer were observed while the next dermal layer showed severe haemorrhages with leucocytic aggregation (Fig 4d). Necrosis and dissociation of muscle fibers, fatty infiltrates and leucocytic infiltration were observed, primarily by lymphocytes, with few numbers of eosinophils between them. Dilatation of lymph vessels (LVs) and leucocytic infiltration were observed, primarily by lymphocytes, as well as few numbers of eosinophils between them (Fig 4e). Severe necrotizing vasculitis was observed in different areas (Fig 4f). Red viral

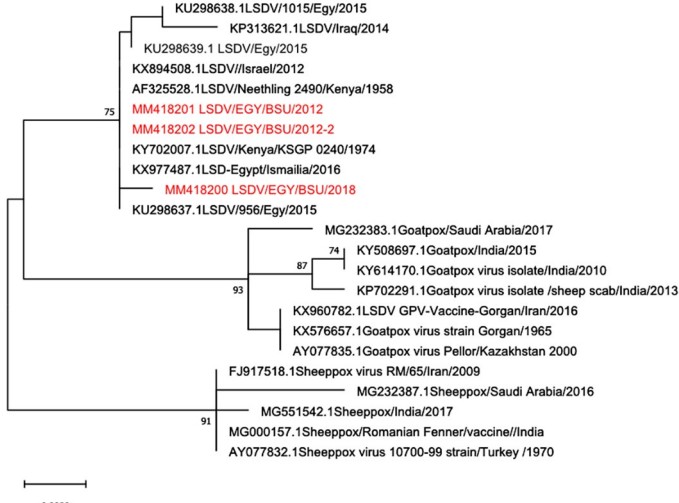

**Fig 5. Phylogenetic analysis of P32 gene.** Tree was generated using Mega-X program by the neighbor-joining analysis. Bootstrap confidence values were calculated on 1000 replicates according to the maximum-likelihood approach.

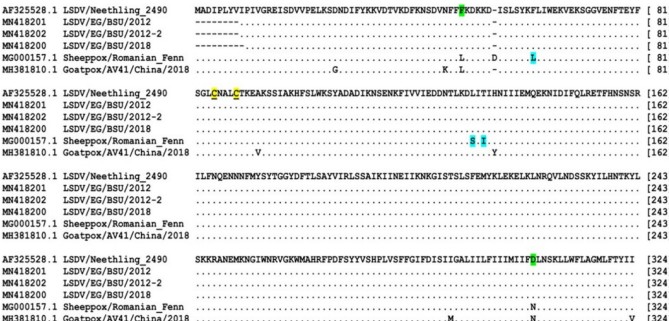

**Fig 6. Deduced amino acid sequence of P32 gene.** Dots indicate identical amino acids, green shaded letters denote unique sequences to LSDV, blue shaded letters denote unique sequences to SPPV and yellow underlined letters denote conserved cysteine residues.

particles were detected in the macrophage of connective tissue of the dermal layer using immunohistochemistry (Fig 4g).

## Discussion

Fever and skin nodules covering the entire skin, lymphadenopathy and oedema were the most significant clinical signs in all suspected cases of LSD during all seasons. This could be explained by the fact that a vector-free season never existed in Egypt, meaning an outbreak could occur all year round and not limited to warm and humid seasons. The existence of different watercourses and water ponds close to cattle populations can also trigger the abundance of the insect vectors, as reported by Tuppurainen (2017). Additionally, the investigated cattle in this study were found to be infested by hard ticks, whose role in LSDV transmission has been previously evaluated [18–21]. A considerable case fatality rate (72/1050, 6.86%) was reported among the examined animals; this rate was higher than that observed in previous

Egyptian outbreaks [5, 6, 22]. Epidemiologically, along the two-year (2017–2018) course of this investigation, LSD has been observed only in cattle, which is still considered the main natural host as reported by [23].

Age (less than 5 months), pregnancy and blood parasites (*Babesia* and *Theileria* species infections) were considered potential risk factors associated with increased fatality. The latter was mostly observed in the Aswan governorate, as it is the most blood parasite enzootic area in Egypt. In addition, some severely affected animals showed severe pneumonia and enteritis. On the other hand, the drastic effect of LSDV may potentially overwhelm the immunity of diseased animals. This supports the presumed immunosuppressive properties of the LSDV, as previously reported [24, 25].

The histopathological findings also revealed serious changes. Haemorrhages and severe destruction of all skin layers in the current study were suggestive of a presumably higher severity of the present 2017–2018 outbreaks. Necrotizing vasculitis of the dermal blood vessels was found to be a major histopathological feature of LSD, as shown in previous studies [26–29].

In this study, 10 (100%) of tested clinical samples were positive for LSDV using PCR (RPO30) and immunohistochemistry. The P32 gene sequences obtained in this study showed no genetic variation from ancestral 1958 LSDV. This finding elucidates the highly conserved nature and the genetic stability of LSDV. Such homology confirmed the highly conserved nature amongst LSDV, SPPV and GTPV [9, 24, 30].

Since 1988, LSD has been persistently reported in Egypt, with severe outbreaks until 2018 despite vaccination campaigns using heterologous vaccine (Romanian sheeppox vaccine). On the basis of current observations and previously reported evidence [31–33], it cannot be ruled out that insufficient and incomplete protection was achieved after heterologous vaccination. On the contrary, other countries successfully experienced vaccination campaign with a homologous LSD vaccine with no further outbreaks were reported [7, 34].

Incomplete vaccination coverage, improper conditions for vaccine storage and transport and the presence of pre-existing immunosuppressive diseases could also be among possible reasons for LSDV outbreaks [34]. There are conflicting results regarding the development of humoral immune response following SPPV vaccine in cattle. No antibody response was detected in cattle following vaccination with SPPV vaccine [35, 36]. However, a partial protection was recorded following challenge with the field strain [35, 36]. In contrast, some studies reported induction of both humoral cell-mediated responses were reported following vaccination of Romanian strain in cattle [37–39]. Humoral antibody response was detected in 2/3 of the vaccinated animals following vaccination with a trivalent vaccine containing SPPV Romania, GTPV and KSGP 0180 [36]. Humoral response was found to be acceptable after using $10^{4.0}$ TCID50 of RM65 SPP vaccine [40]. Interestingly, it was found that in spite of vaccination, the local immune evasion strategy adopted by capripoxviruses locally alleviates the host's immune response to allow the virus to replicate at the site of entry [41].

In conclusion, the mass vaccination campaigns in Egypt that were based on the use of a heterologous vaccine (Romanian sheeppox vaccine) provided insufficient levels of protection. This necessitates the use of an alternative strategy in utilizing the homologous LSDV vaccine. A strategic plan is strongly recommended, particularly one that ensures mass coverage of the vaccination among cattle populations all over Egypt.

## Supporting information

**S1 Raw images.**
(PDF)

## Author Contributions

**Conceptualization:** Aml M. Abdel- Ra'ouf, Bahaa S. Madkour, Hosein I. Hosein.

**Data curation:** Aml M. Abdel- Ra'ouf, Bahaa S. Madkour.

**Methodology:** Sherin R. Rouby, Nesreen M. Safwat, Khaled H. Hussein.

**Supervision:** Ahmed S. Abdel-Moneim, Hosein I. Hosein.

**Validation:** Ahmed S. Abdel-Moneim.

**Visualization:** Bahaa S. Madkour.

**Writing – original draft:** Sherin R. Rouby, Ahmed S. Abdel-Moneim, Hosein I. Hosein.

**Writing – review & editing:** Ahmed S. Abdel-Moneim, Hosein I. Hosein.

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
