## [Decision Letter · Decision Letter 0]

2 Sep 2021

PONE-D-21-20980

Lumpy skin disease outbreaks in Egypt during 2017-2018 among sheeppox vaccinated cattle: Epidemiological, pathological, and molecular findings

PLOS ONE

Dear Dr. Rouby,

Thank you for submitting your manuscript to PLOS ONE. After careful consideration, we feel that it has merit but does not fully meet PLOS ONE’s publication criteria as it currently stands. Therefore, we invite you to submit a revised version of the manuscript that addresses the points raised during the review process.

Many thanks for submitting your manuscript to PLOS One

It was reviewed by two experts in the field, and they have recommended some modifications be made prior to acceptance

I therefore invite you to make these changes and to write a response to reviewers which will expedite revision upon resubmission

I wish you the best of luck with your modifications

Hope you are keeping safe and well in these difficult times

Thanks

Simon

We look forward to receiving your revised manuscript.

Kind regards,

Simon Clegg, PhD

Academic Editor

PLOS ONE

2. You indicated that you had ethical approval for your study. In your Methods section, please ensure you have also stated whether you obtained consent from farmers whose animals were included in the study or whether the research ethics committee or IRB specifically waived the need for their consent.

Reviewers' comments:

Reviewer's Responses to Questions

**Comments to the Author**

1. Is the manuscript technically sound, and do the data support the conclusions?

Reviewer #1: Partly

Reviewer #2: Yes

2. Has the statistical analysis been performed appropriately and rigorously? 

Reviewer #1: N/A

Reviewer #2: Yes

3. Have the authors made all data underlying the findings in their manuscript fully available?

Reviewer #1: Yes

Reviewer #2: Yes

4. Is the manuscript presented in an intelligible fashion and written in standard English?

Reviewer #1: Yes

Reviewer #2: Yes

5. Review Comments to the Author

Reviewer #1: The authors describe the substantial outbreak of LSD in SPPV-vaccinated cattle in Egypt.

In general, the manuscript is easy to read and the presented data support the conclusions. Nevertheless, some additional information would be improve the manuscript. If the according information are not available, it should be clearly stated out.

Major comments:

• In total, 1050 clinically affected cattle were screened, but it is not clear what susceptible cattle population is available in the Beni-Suef, Sohag and Aswan Governorates and how many of the cattle were vaccinated with the SPPV vaccine.

• Furthermore, it is not clear for me, at what time the SPPV vaccination was performed and how the vaccination procedure was recorded for each individual cattle.

• In this context, it would be helpful to know, how many of the verified SPPV vaccinated cattle produced an immunological response against the SPPV vaccine (humoral and/or cellular). If such data are not available for these cattle population, the authors should screen the literature for according data.

• It is not clear if the Egypt SPPV Romanian vaccine produce no immune response in cattle, in general. Alternatively, the heterologous immune response based on the SPPV vaccine could be not sufficient to protect for the LSDV infection. This problem should be discussed more intensive.

• An interesting study would be the analysis of the serological response after the SPV vaccination. Here, cattle from the field should be tested before and 6-8 weeks after vaccination to evaluate the rate of successful immunised cattle. But this experiment can be also the part of one of the next studies.

Minor comments:

• Page3, line 9: a dot after 4] is missing

• P3, l21: RPO-30

• P7, l12: RPO-30

• P7, l14: Fig. 4G is described here at the first time, thus the legend Fig. 4A would be better.

• P7, l18: Why only one of 10 PCR amplicons for the p32 sequence showed readable sequences. A confirmation of the p32 sequence for other LSDV-field strains from 2017/18 would be more robust for the conclusion.

Reviewer #2: This is a nice and interesting article which is simple to read and well presented which described LSDV in Egypt.

I have made a few comments- line numbers start from the first text line of the intro as there were none on the submitted manuscript.

It would be nice to know some figures for the areas- number of cows, % vaccinated etc, and how recent the vaccination was for the cattle.

Also, is it that the vaccination hasn’t produced a response? Or is it some other reason why the animal has become diseased?

A few minor comments are below.

Line 1- comma after devastating

Line 3- delete a

Line 6- reword to ..related to the genus ….

Line 24- I may be incorrect, but isn’t this the RPO-30? Please check throughout

Line 27 and 28- this is repetitive, please modify

Line 38- delete the second ‘of’

Line 63- please include full PCR reagents

Please ensure that all reagents have the manufacturer in the methods section

Line 77- remove brackets from dilution

Line 78- please define AP

P32 PCR- please include PCR reagents

Line 107- how were these samples chosen?

Line 117- please use similarities when not 100% identical

Line 148- comma after LSDV

Line 159-160= you don’t show this data- unless I missed it, and this is quite important

Not sure that figures 3 or 6 are needed but up to the authors

6. PLOS authors have the option to publish the peer review history of their article (what does this mean?). If published, this will include your full peer review and any attached files.

Reviewer #1: No

Reviewer #2: No

---

## [Author Response · Author response to Decision Letter 0]

23 Sep 2021

I'd like to thank you for your valuable suggestions. All recommendations will be considered. 

Reviewer 1:

Q1: In total, 1050 clinically affected cattle were screened, but it is not clear what susceptible cattle population is available in the Beni-Suef, Sohag and Aswan Governorates and how many of the cattle were vaccinated with the SPPV vaccine. 

Response: 

Thank you for your kind comment, the study only concerned with the animals showing the typical signs of the disease. The numbers of suspected animals in each governorate are shown in and under figure 1. Regarding the susceptibility of cattle and the number of vaccinated cattle, it was assumed that these animals are not susceptible because vaccination in cattle is compulsory all over the country where all animals are vaccinated regularly every 6 months since years. However, disease outbreaks still occur.

Q2: Furthermore, it is not clear for me, at what time the SPPV vaccination was performed and how the vaccination procedure was recorded for each individual cattle.

Response: 

Thank you for your kind comment, all animals in this investigation were vaccinated (3 months) prior to developing the disease. As mentioned in method section “All affected animals were vaccinated within the national annual vaccination program with the Romanian SPPV vaccine (103 TCID50, Veterinary Serum and Vaccine Research Institute [VSVRI], Egypt). The development of the disease was reported three months after vaccination.”

Q3: In this context, it would be helpful to know, how many of the verified SPPV vaccinated cattle produced an immunological response against the SPPV vaccine (humoral and/or cellular). If such data are not available for these cattle population, the authors should screen the literature for according data.

-It is not clear if the Egypt SPPV Romanian vaccine produce no immune response in cattle, in general. Alternatively, the heterologous immune response based on the SPPV vaccine could be not sufficient to protect for the LSDV infection. This problem should be discussed more intensive.

-An interesting study would be the analysis of the serological response after the SPV vaccination. Here, cattle from the field should be tested before and 6-8 weeks after vaccination to evaluate the rate of successful immunised cattle. But this experiment can be also the part of one of the next studies.

Response: 

Thank you for your kind comment, animals were not assessed for their immunological response. We have added a paragraph regarding the immunological response to SPPV in cattle.

Q4: It is not clear if the Egypt SPPV Romanian vaccine produce no immune response in cattle, in general. Alternatively, the heterologous immune response based on the SPPV vaccine could be not sufficient to protect for the LSDV infection. This problem should be discussed more intensive.

Response: 

This is the fact in Egypt and therefore it was an important purpose to perform this study. This was stated in the manuscript: (Since 1988, LSD has been persistently reported in Egypt, with severe outbreaks until 2018 despite vaccination campaigns using heterologous vaccine (Romanian sheeppox vaccine). In addition, we have added a paragraph regarding the immunological response to SPPV in cattle. 

Minor comments:

• Page3, line 9: a dot after 4] is missing

• P3, l21: RPO-30

• P7, l12: RPO-30

• P7, l14: Fig. 4G is described here at the first time, thus the legend Fig. 4A would be better.

• P7, l18: Why only one of 10 PCR amplicons for the p32 sequence showed readable sequences. A confirmation of the p32 sequence for other LSDV-field strains from 2017/18 would be more robust for the conclusion.

Response: 

All were changed as recommended

P7, l14: Fig. 4G: it was changed to Fig. 4g not A as it is the best figure confirm the presence of a viral antigen “Describe the Red viral particles in macrophage of connective tissue of the dermal layer (black arrows) using alkaline phosphatase immunohistochemistry”

-Why only one of 10 PCR amplicons for the p32 sequence showed readable sequences. A confirmation of the p32 sequence for other LSDV-field strains from 2017/18 would be more robust for the conclusion.

Response: 

Sequences was done to prove the stability of LSDV DNA by comparing the current circulating strain with 2012 LSDV strain. Results confirm the stability of LSDV DNA and came in accordance with that mentioned by Tuppurainen et al., 2017 “ There is only one serological type of LSDV, The large, double-stranded DNA virus is very stable, and very little genetic variability occurs. Therefore, for LSDV, farm-to-farm spread cannot be followed by sequencing the virus isolates, as is done with other TADs, e.g. foot-and-mouth disease (FMD).

Tuppurainen, E., Alexandrov, T. & Beltrán-Alcrudo, D. 2017. Lumpy skin disease field manual –

A manual for veterinarians. FAO Animal Production and Health Manual No. 20. Rome. Food and Agriculture Organization of the United Nations (FAO). 60 pages.

Reviewer: 2: 

Reviewer #2: This is a nice and interesting article which is simple to read and well-presented which described LSDV in Egypt.

I have made a few comments- line numbers start from the first text line of the intro as there were none on the submitted manuscript.

It would be nice to know some figures for the areas- number of cows, % vaccinated etc, and how recent the vaccination was for the cattle.

Thank you for your kind comment, such figures are available in the Map (Fig. 1) and Table 1. All animals are vaccinated (Mandatory vaccination). Vaccination in this study as cited was 3 months prior to developing of the disease.

Also, is it that the vaccination hasn’t produced a response? Or is it some other reason why the animal has become diseased?

Insufficient immunity produced but this was not a target in this study.

The drastic effect of the virus. 

The presence of pre-existing immunosuppressive diseases.

Pregnancy and blood parasites (Babesia and Theileria species infections), pneumonia and enteritis were considered potential risk factors associated with increased fatality.

A few minor comments are below.

Line 1- comma after devastating

Line 3- delete a

Line 6- reword to ..related to the genus ….

Line 24- I may be incorrect, but isn’t this the RPO-30? Please check throughout

Line 27 and 28- this is repetitive, please modify

Line 38- delete the second ‘of’

Line 63- please include full PCR reagents

Please ensure that all reagents have the manufacturer in the methods section

Line 77- remove brackets from dilution

Line 78- please define AP

P32 PCR- please include PCR reagents

Line 107- how were these samples chosen?

Line 117- please use similarities when not 100% identical

Line 148- comma after LSDV

Line 159-160= you don’t show this data- unless I missed it, and this is quite important

Not sure that figures 3 or 6 are needed but up to the authors

Response: thank you for your valuable suggestions. All recommendations will be considered. 

I have attached a copy that highlights changes made to the original version

You indicated that you had ethical approval for your study. In your Methods section, please ensure you have also stated whether you obtained consent from farmers whose animals were included in the study or whether the research ethics committee or IRB specifically waived the need for their consent.

Response: 

It was added as recommended “The animal ethical committee of the Faculty of Veterinary Medicine, Beni-Suef University, Egypt, approved the present study. Clinical samples used in this study were collected after approval of all the animals’ owners.”

Study’s minimal underlying data

Accession numbers

MN418201 

MN418202 

MN418200

---

## [Editor Report · Decision Letter 1]

5 Oct 2021

Lumpy skin disease outbreaks in Egypt during 2017-2018 among sheeppox vaccinated cattle: Epidemiological, pathological, and molecular findings

PONE-D-21-20980R1

Dear Dr. Rouby,

We’re pleased to inform you that your manuscript has been judged scientifically suitable for publication and will be formally accepted for publication once it meets all outstanding technical requirements.

Kind regards,

Simon Clegg, PhD

Academic Editor

PLOS ONE

Additional Editor Comments:

Many thanks for resubmitting your manuscript to PLOS One

As you have addressed all the comments and the manuscript reads well, I have recommended it for publication

You should hear from the Editorial Office shortly.

It was a pleasure working with you and I wish you the best of luck for your future research

Hope you are keeping safe and well in these difficult times

Thanks

Simon

---

## [Editor Report · Acceptance letter]

12 Oct 2021

PONE-D-21-20980R1 

Lumpy skin disease outbreaks in Egypt during 2017-2018 among sheeppox vaccinated cattle: Epidemiological, pathological, and molecular findings 

Dear Dr. Rouby:

I'm pleased to inform you that your manuscript has been deemed suitable for publication in PLOS ONE. Congratulations! Your manuscript is now with our production department. 

Kind regards, 

on behalf of

Dr. Simon Clegg 

Academic Editor

PLOS ONE